# Biochemical analysis, photosynthetic gene (psbA) down–regulation, and *in silico* receptor prediction in weeds in response to exogenous application of phenolic acids and their analogs

**Sobia Anwar, Saadia Naseem, Zahid Ali***

Department of Biosciences, Plant Biotechnology and Molecular Pharming Laboratory, COMSATS University Islamabad (CUI), Islamabad, Pakistan

* zahidali@comsats.edu.pk

**Data Availability Statement:** All relevant data are within the paper and its Supporting Information files.

## Abstract

Chemical herbicides are the primary weed management tool, although several incidences of herbicide resistance have emerged, causing serious threat to agricultural sustainability. Plant derived phenolic acids with herbicidal potential provide organic and eco-friendly substitute to such harmful chemicals. In present study, phytotoxicity of two phenolic compounds, ferulic acid (FA) and gallic acid (GA), was evaluated *in vitro* and *in vivo* against three prevalent herbicide-resistant weed species (*Sinapis arvensis*, *Lolium multiflorum* and *Parthenium hysterophorus*). FA and GA not only suppressed the weed germination (80 to 60% respectively), but also negatively affected biochemical and photosynthetic pathway of weeds. In addition to significantly lowering the total protein and chlorophyll contents of the targeted weed species, the application of FA and GA treatments increased levels of antioxidant enzymes and lipid peroxidation. Photosynthetic gene (psbA) expression was downregulated (10 to 30 folds) post 48 h of phenolic application. *In silico* analysis for receptor identification of FA and GA in psbA protein (D1) showed histidine (his-198) and threonine (thr-286) as novel receptors of FA and GA. These two receptors differ from the D1 amino acid receptors which have previously been identified (serine-264 and histidine-215) in response to PSII inhibitor herbicides. Based on its toxicity responses, structural analogs of FA were also designed. Four out of twelve analogs (0.25 mM) significantly inhibited weed germination (30 to 40%) while enhancing their oxidative stress. These results are unique which provide fundamental evidence of phytotoxicity of FA and GA and their analogs to develop cutting-edge plant based bio-herbicides formulation in future.

## Introduction

The use of chemical herbicides has considerably controlled the weeds thereby facilitating agricultural growth globally [1]. Increasing incidents of herbicide resistance in weeds has however hampered their affectivity [2–4]. Since the first incidence of resistance development was

**Funding:** This research was supported by Pakistan Science Foundation (PSF) under project no. PSF-NSLP-663. The funders had no role in study design, data collection and analysis, decision to publish, or preparation of the manuscript.

**Competing interests:** The authors have declared that no competing interests exist.

recorded in 1957, weeds of 96 crops in 72 countries have developed resistance to 21 out of 31 classes of herbicides [5]. Weeds tend to develop resistance to herbicides when applied repeatedly with the same mode of action [6]. According to active ingredients, atrazine has been documented to develop resistance in the highest number of weed resistant species [7], followed by imazethapyr [8], tribenuron-methyl [9], imazamox [10], chlorsulfuron [11] metsulfuron-methyl [12], glyphosate [13], iodosulfuron-methyl-sodium [14], fenoxaprop-P-ethyl [15], simazine [15], bensulfuron-methyl [16], thifensulfuron-methyl [17], fluazifop-P-bityl [18] and pyrazosulfuron-ethyl [5, 18]. The most prevalent herbicide-resistant biotypes, which have lowered agricultural production by 40% to 90%, include *S. arvensis*, *L. multiflorum*, *and P. hysterophorus* [19–24]. Photosystem II inhibitor, fatty acid inhibitor, synthetic auxin inhibitor, acetolactate synthase (ALS), acetyl CoA carboxylase (ACCase), and 5-enolpyruvylshikimate-3-phosphate synthase (EPSPS) resistance has been reported in *S. arvensis* (family *Brassicaceae*), *P. hysterophorus* (family *Asteraceae*) and *L. multiflorum* (family *Poaceae*) [5, 22, 25].

In light of the significant challenges posed by already existing synthetic herbicides and resistant weeds, substitute to harmful chemicals with potentially varied modes of action is necessary for long-term weed control strategies. Employment of plant based allelochemicals in this regard could progress towards greener and safer agriculture providing wide range of opportunities to cope with weed management [26–28]. The allelochemicals are secondary metabolites generated by different plant parts as a result of exudation, volatilization, and leaching and they have the potential to affect the development of nearby plants [29]. Different environmental factors significantly affect the availability, biochemistry, and toxicity of these phytochemicals when these are released into the rhizosphere [30]. Considerations for the rhizosphere include microbes, nearby plants, organic and inorganic materials, pH, consistency, and moisture content [31]. Prior to reaching the target species, these conditions may have an impact on the physiochemical nature of allelopathins in the rhizosphere and consequently their phytotoxic activity [32–34]. The required concentration of potent allelopathins must therefore be examined under edaphic scenario in order to precisely determine their phytotoxicity [35].

Allelopathins, made up of phenolic acids, flavonoids, alkaloids, steroids, and fatty acids, cause phytotoxicity in weeds by interfering with their photosynthetic system, metabolic enzymes, mitochondrial respiration, cell division, lignin content, and protein synthesis, causing oxidative stress and inhibiting weed growth [18, 36, 37]. Phenolic compounds are a group of plant allelochemicals that are both significant and common in the ecosystem. These are chemical compounds with an aromatic hydrocarbon group linked straight to a hydroxyl group (-OH) [17]. Due to their antioxidant characteristics, these chemicals are particularly effective as allelopathins and work on various sites of cellular processes [38], presenting a variety of phytotoxins that can be employed as the basis for "green" herbicides, so called as "bioherbicides." Bioherbicides are naturally derived products such as pathogens and other microorganisms, as well as microbes, insects, or plant extracts based phytotoxins, that operate as a natural weed control method [16, 39].

Allelopathicity of phenolics has been thoroughly studied and documented, indicating that these metabolites have the capability to obstruct critical physiological functions in plants by interacting with membrane permeability, organic compound production, hormone activity, photosynthesis, and respiration [17]. Gallic acid (GA) and ferulic acid (FA), the dehydroshikimic acid derivatives, are the most frequent phenolic acids found in hydrolysable tannins [15]. Both FA and GA significantly reduce leaf moisture content, root development and elongation and nutrient absorption, preventing weeds from germinating and spreading [17]. Photosynthetic activity of weeds is also reported to be inhibited by phenolic acids such as p-hydroxy benzoic acid and FA [40]. Furthermore, these allelochemicals are soluble in aqueous solution, so they do not require the addition of surfactants to be applied [14]. Instead of heavy molecules present in herbicides that stick to the soil, these plant-based chemicals have more hydrogen

and oxygen, therefore they have a short half-life and thus do not accumulate in soil, making them eco-friendly [13].

Despite the fact that the allelopathic activity of both FA and GA has been studied previously [10–12, 41], their phytotoxic effect on physiological and biochemical mechanisms of major herbicide-resistant weeds such as *S. arvensis*, *L. multiflorum*, and *P. hysterophorus* must be studied under rhizospheric conditions. Therefore, the purpose of this study was to determine the phytotoxic potential of FA and GA against the physiological and biochemical processes of weed seedling growth, photosynthetic pigment and protein content, and enzymatic and non-enzymatic activities of weeds, as well as the *in silico* identification of their specific binding receptors in D1 protein of PSII. Since phytotoxic chemicals are produced in small amounts in plants and their extraction in sufficient amount and pure form is challenging [42], thus chemometric tools were used to design structural analogues of FA to investigate and compare their herbicidal activity with natural compounds. This study will help to discover the varied metabolic pathways that could be utilized to discover allelochemicals unique mode of actions for sustainable management of weeds in agriculture.

## Material and methods

Phytotoxic effect of ferulic acid (FA) and gallic acid (GA) and their structural analogs was evaluated against prevalent and resistant weeds (*S. arvensis*, *L. multiflorum* and *P. hysterophorus*). Research work was performed at Plant Biotechnology and Molecular Pharming (PBMP) Laboratory, Department of Biosciences, COMSATS University Islamabad, Pakistan.

### *In vitro* weed seed germination bioassay with FA and GA

Weed seeds were acquired from Department of Weed Science, University of Peshawar, Pakistan. Analytical grade phenolic acids, FA and GA were purchased from Sigma Aldrich, USA and were weighed accurately (1.9 and 1.7 g respectively) to prepare 10 mM stock solutions. The chemicals were dissolved separately in methanol and water (1:9), later methanol was evaporated and molarity of solutions was adjusted to 3 mM by diluting them. Solutions were filter sterilized with 0.45 μm syringe filter and stored at 4 ˚C after maintaining their pH (6.0) with sodium hydroxide [40]. Preliminary investigation for FA and GA dose optimization was conducted against germination of a dicot (*S. arvensis*) and a monocot (*L. multiflorum*) weed respectively. Both the phenolic compounds were individually added in 1.5 mL tissue culture medium (MS plus salts = 4.4 gL$^{-1}$, agar = 8 gL$^{-1}$, pH 5.8) contained in glass tubes (60 mL, 140 × 27.7 mm, CNW) to make their final concentration to 0.6, 0.8 and 1 mM for each weed. MS-agar medium without FA and GA was used as a control. Seeds were surface sterilized with 5% sodium hypochlorite, followed by thorough washing with distilled water and drying on filter paper [43]. The seeds were inoculated in individual glass tubes (four seeds per tube). Tubes were covered with screw lids and placed in growth room with 16/8 h light/dark photoperiod at 25 ˚C. Seed germination was recorded after 3 and 5 days for *S. arvensis* and *L. multiflorum* respectively, to calculate their germination index and germination percentage. There were ten replicates of each treatment and the experiment was repeated three times.

### *In vivo* phytotoxicity of FA and GA against growth of *Sinapis arvensis*, *Lolium multiflorum* and *Parthenium hysterophorus*

Sterilized weed seeds (10 to 15) were sown in pots (7.5 by 7.8 cm) with 100 g soil and compost mixed in equal proportion. Pots were placed in a growth chamber under natural conditions at 25 ˚C temperature and 16/8 h light/dark photoperiod. Each pot received distilled water regularly to maintain soil moisture. Thereafter, uniform sized and healthy seedlings, of *P.*

*hysterophorus*, *L. multiflorum* and *S. arvensis* (at least 10 to 15) were kept in containers (7.5 by 7.8 cm with 100 g manure soil mixture) under same growth conditions. Two weeks old stable seedlings were used for experimental procedure.

During preliminary phytotoxicity investigation of FA and GA in soil, against target weeds, lower concentrations i.e. 0.4, 0.6 and 0.8 mM solutions could not induce any effect on their growth; hence, 1, 2 and 3 mM of each compound was used for further analysis. 1 mL of each treatment was applied individually with pipette to the seedlings in all pots. Same amount of distilled water was applied as a control. Treatments were applied six times after every 24 h with three replicates of each treatment. After 48 h post last treatment, seedlings from each pot were used to measure plant growth and biomass. Radicle and hypocotyl lengths (mm) were measured with scale while dry biomass (mg) was recorded with electronic weighing balance. Experiment was performed thrice in a completely randomized design.

## Total chlorophyll and protein content assessment of weeds

Total chlorophyll content (chlorophyll a and b) was calculated from harvested seedlings, as described by [44]. Weed leaves from each treatment were lyophilized in liquid nitrogen. Freeze dried leaves (0.5 g) were dissolved in 1 mL dimethyl sulfoxide (DMSO). Mixtures were incubated at 60 ˚C for 30 min. Absorbance was measured with spectrophotometer (SPECORD 50) at 645 nm and 663 nm.

Protein content of the collected leaves was determined following the method of [45]. Fresh leaves samples (0.1 g) were ground in a pestle and mortar with 1 mL of phosphate buffer (pH 7.5) and centrifuged at 3000 rpm for 12 min. The supernatant was transferred to 15 mL falcon tubes, followed by addition of 1 mL distilled water and 1 mL of reagent C (alkaline copper solution). Solution was mixed thoroughly for 10 min and then 1 mL of the reagent D (copper carbonate solution) was added. The absorbance of samples was observed at 650 nm after the incubation of 30 min. The concentration of soluble protein content was determined spectrophotometrically with reference to standard curve prepared with bovine serum albumin. All measurements were taken in triplicates.

## Antioxidant enzymes activity of weeds against FA and GA treatments

Phytotoxic effect of FA and GA on biochemical pathways of weeds was analyzed enzymatically and non-enzymatically. For enzyme extraction, 50 mg of each weed sample was lyophilized and homogenized in phosphate buffer saline (50 mM, pH 7.0), followed by centrifugation at 10,000 g for 20 min at 4 ˚C. Supernatant was transferred to 1.5 mL Eppendorf tubes for and stored at –20 ˚C for further analysis.

Catalase (CAT) activity was estimated as previously described by [9] using $H_2O_2$ as substrate. Observations were obtained spectrophotometrically at 240 nm and results were expressed as mM of $H_2O_2$ oxidized $min^{-1}$ $g^{-1}$ FW (fresh weight of plant). Superoxide dismutase (SOD) assay was performed as described by [8]. SOD activity was calculated by determining the inhibition of nitroblue tetrazolium chloride (NBT) photoreduction at 560 nm and results were expressed as mM $min^{-1}$ $g^{-1}$ FW. Further, peroxidase (POD) activity was determined by measuring guaicol oxidation to tetraguaicol at 470 nm, following methodology of [46]. POD unit was calculated as mM $min^{-1}$ $g^{-1}$ FW.

## Estimation of weeds lipid peroxidation

Malondialdehyde (MDA) concentration was determined to assess lipid peroxidation in weed samples by method illustrated by [47]. Homogenates were prepared with 0.1 g leaf tissue and 1.0% trichloroacetic acid (TCA). Supernatant was collected after centrifugation at 10,000 g for

15 min at 4 °C. Absorbance was measured at 532 nm and 600 nm after mixing 2 mL thiobarbituric acid (TBA) (0.5% in 20% TCA) and incubating at 95 °C for 15 min. Reaction was stopped by sample incubation on ice for 10 min prior to absorbance measurement. Results were expressed as mM $g^{-1}$ FW.

## Expression analysis of photosynthetic gene (psbA) of weeds in response to FA and GA treatments

Weed seedlings treated with FA and GA were harvested after 48 h of last treatment and were analyzed for psbA gene expression through semi-quantitative reverse transcriptase polymerase chain reaction (RT-PCR). Total RNA was isolated from all samples using Purelink RNA mini-kit (Invitrogen, Thermoscientific kit), according to the protocol. RNA was quantified with Colibro microvolume spectrophotometer (Titertek, Brethhold). cDNA synthesis and PCR analysis was carried out from equal amount of total RNA (1 μg) from each sample utilizing One-Step kit (Invitrogen Life Technologies, Carlsbad, CA, USA). Primers for target gene were designed using OligoAnalyzer 3.1 software, and by alignment of the nucleotide sequences from species representing several plant families available in GenBank (National Center for Biotechnology Information). Target sites for designed primers were determined with snapgene viewer 4.2.11 software. Tubulin was used as internal control. Primer sequences (psbA and tubulin) were as follows: psbA F; AGCTCCTGTTGC AGCTGCTACT, psbA R; GCCGAATACA CCAGCTACACCTAA and tubulin F; GGTAACATTGTGCTCAGTGGTGG, tubulin R; AACGA CCTTAATCTTCATGCTGC with annealing temperatures of 64 °C and 58 °C respectively.

The PCR mix contained 2 μL of the template from each sample, 1 μM dNTPs, 0.4 μM primers (forward and reverse), 0.12 μL of Taq polymerase (Invitrogen) and 10X 1 μL PCR buffer (Invitrogen) in a final volume of 20 μL. RT-PCR conditions were 50 °C for 30 min and 94 °C for 2 min, then, 94 °C for 15 s, 55 °C for 30 s and 70 °C for 1 min for 35 cycles, and final extension 72 °C for 10 min. PCR products were analyzed on 1% (w/w) agarose gel containing 0.5 μg $mL^{-1}$ ethidium bromide. Equal amount of each sample (5 μL) was carefully mixed and loaded with loading dye (2 μL) to each well to determine the accurate differences between gene expressions in samples from each treatment. Gels were visualized and photographed for analysis in gel documentation system (Genosens 1560). PCR amplifications were carried out at least two times individually. ImageJ software was used to measure percentage area intensities of bands for quantification of gene expression levels [48]. Data were normalized to reference gene for accurate determination of psbA gene expression.

## Docking analysis of FA and GA with psbA (D1) protein

D1 protein of photosynthetic gene (psbA) was selected as a target receptor for docking studies. The protein (D1) sequences of *S. arvensis*, *L. multiflorum* and *P. hysterophorus* were taken from NCBI. Bioedit was used for homology modeling of query sequences with D1 sequence of *Arabidopsis thaliana*, showing >99% homology. Therefore, protein data bank (PDB) model of D1 protein of *A. thaliana*, obtained from Research Collaboratory for Structural Bioinformatics (RCSB) repository, was used for further analysis in Schrodinger software. All of the water molecules and hydrophobic chains were removed prior with Protein Preparation Wizard [49].

2D molecular structures of ferulic acid and gallic acid (referred as ligands afterwards) were obtained as sdf file from Pubchem database (https://pubchem.ncbi.nlm.nih.gov). Ligands were prepared by adding hydrogen bonds and filling in missing side chains with LigPrep. Glide application was used for generation of receptor grid by reducing its size and energy in Maestro and then docking to proteins. Glide scores [empirical score that depicts ligand binding energy; [50] were used for evaluating docked conformers. Docked images were generated with PyMOL.

## Structural analog designing

Due to higher weed growth inhibitory effects, ferulic acid was used as a base compound to design its structural analogs. By high-throughput screening (HTS) collection library of enamine database, several compounds (1,834,363) were recovered and annotated with OpenEye software tools. Elimination of unnecessary compounds was performed with filter application in OpenEye software (https://www.eyesopen.com) before proceeding to virtual screening. Three dimensional multi-conformers were generated with OMEGA software after removing unwanted molecules. At most 200 conformers were produced with root mean square distance (RMS = 0.5 Å) and E-window (10 kcal mol$^{-1}$). Charges were assigned to conformers with Molecular Force Field (MMFF). Virtual rapid overlay of chemical structures (vROCS) graphical interface was utilized to generate shape and color (hydrogen bond acceptor, donor and ring) based on reported query allelochemicals. Generated query was used to run ROCS3 (Rapid Overlay of Chemical Structures) code for carrying out screening of prepared chemical database. Tanimoto combo scores were utilized to rank screened compounds yielded on basis of shape and color matching of aligned query molecule. Higher scores represent better match of analogs with FA [51].

## Comparative analysis of toxicity of FA and its analogs against germination and biochemical activity of weeds

*In vitro* interactive experiments on *S. arvensis* and *L. multiflorum* seed germination were performed with all analogs to investigate their toxic effects. Prior to that, 10 mg of each compound (analogs) was dissolved in 5 mL of $10^{-2}$ M dimethsuloxide (DMSO) to make their stock solutions. Each stock solution was diluted to prepare 1 mM working solution. Weed seeds were surface sterilized with 5% sodium hypochlorite (NaOCl) following three washings with autoclaved de-ionized water. Seeds were thoroughly dried on filter paper in laminar flow. Afterwards, they were treated with final concentration of 0.1, 0.25 and 0.5 mM of each of analog added in MS-agar medium contained in glass vials (60 mL, 140 by 27.7 mm, CNW).

For assessing biochemical activity of weeds, post analogs treatment, through CAT, SOD, POD enzymes and lipid peroxidation, pre-germinated seedlings were transferred to MS-agar medium with 0.25 mM analogs concentration. Same dilutions of DMSO and MS-agar without any treatment were used as a negative control while 0.8 mM and 1 mM FA were used for comparative assessment. Treatment values were normalized to 0.25 mM DMSO.

## Statistical analysis

The experiments were conducted in completely randomized design (CRD) and each experiment was repeated three times. Variables were transformed with arcsine and log transformation where required for normalization [37]. Mean values of treatments were compared with multivariate two-way analysis of variance (ANOVA) using generalized linear model. Data were presented as mean ± standard errors of means (SE). Significance of obtained results was analyzed with Tukey's HSD test at probability level $P < 0.05$ with SPSS v. 23 (IBM, USA).

## Results

### *In vitro* weed seed germination bioassay with FA and GA

The significant inhibitory effects of FA and GA application were recorded on seed germination of *S. arvensis* and *L. multiflorum*. *In vitro* germination of *S. arvensis* reduced up to 40 to 60% with 0.8 and 1 mM FA, respectively, while same treatments reduced 60 to 80% germination of *L. multiflorum* (Fig 1). When treated with 0.8 and 1mM GA, 20 to 40% and 50 to 60% germination inhibition was observed in both the weed species, respectively (S1 and S2 Figs). Lower

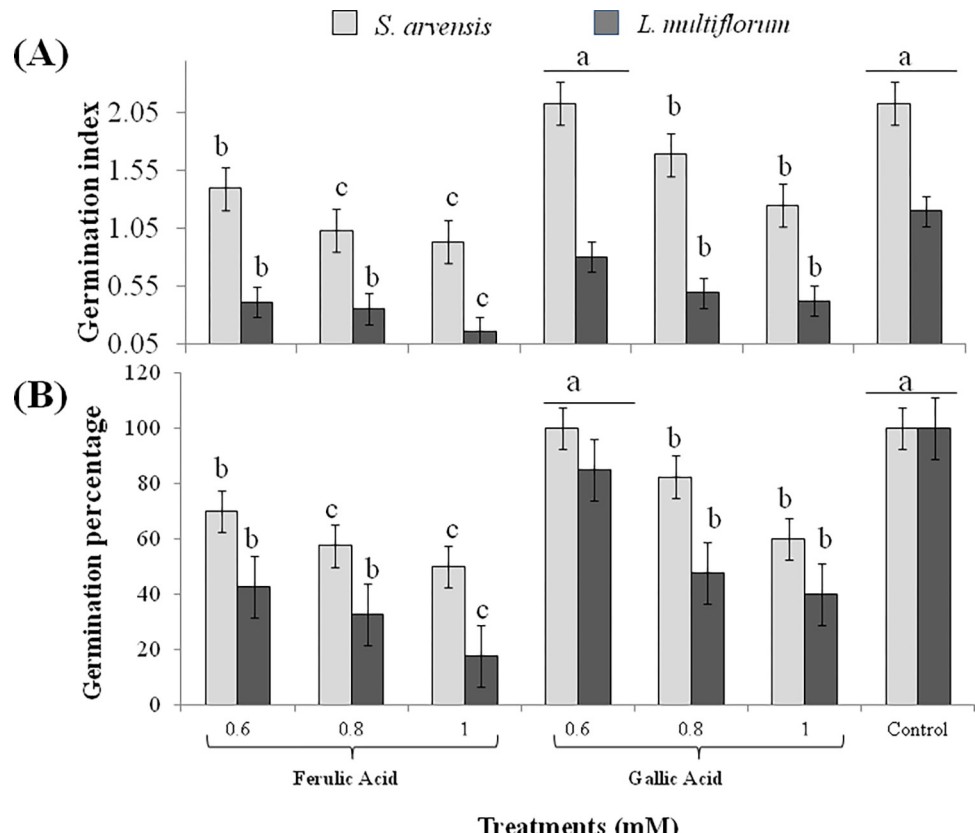

**Fig 1. Growth response of *Sinapis arvensis* and *Lolium multiflorum* to 0.6, 0.8 and 1 mM ferulic acid (FA) and gallic acid (GA) and aqueous control.** (A) Germination index and (B) germination percentage. Data are presented as mean ± SE (n = 10). Bar values with different letters differ significantly at $P < 0.05$.

molarity of FA and GA (0.6 mM) did not show significant effect on germination index and germination percentage of *S. arvensis* and *L. multiflorum*. It was observed that FA had higher inhibitory effect than GA, and *L. multiflorum* was found more susceptible to these treatments as compared to *S. arvensis*.

## *In vivo* phytotoxicity of FA and GA against weed growth

*In vivo* herbicidal effects of FA and GA were observed at 1, 2 and 3 mM concentrations against the seedling growth of *S. arvensis*, *L. multiflorum* and *P. hysterophorus*. The growth of all weeds was significantly affected and permanent wilting was observed after six treatment applications of FA and GA. In comparison to control, all weeds treated with 2 and 3 mM concentration of both compounds suppressed radicle and hypocotyl length as well as reduced the dry biomass (Fig 2). Lower concentration (1 mM) of FA and GA slightly reduced the growth of *S. arvensis* and *P. hysterophorus* but did not affect seedling length and biomass of *L. multiflorum*. FA induced more pronounced effect in comparison to GA.

## Total chlorophyll and protein content assessment of weeds

Total chlorophyll content declined up to 60 to 90% in *S. arvensis* with 3 mM concentration of GA and FA, 60 to 70% in *L. multiflorum* while 50 to 70% in *P. hysterophorus*, respectively. The 1 mM concentration of both compounds did not affect photosynthetic pigment in comparison

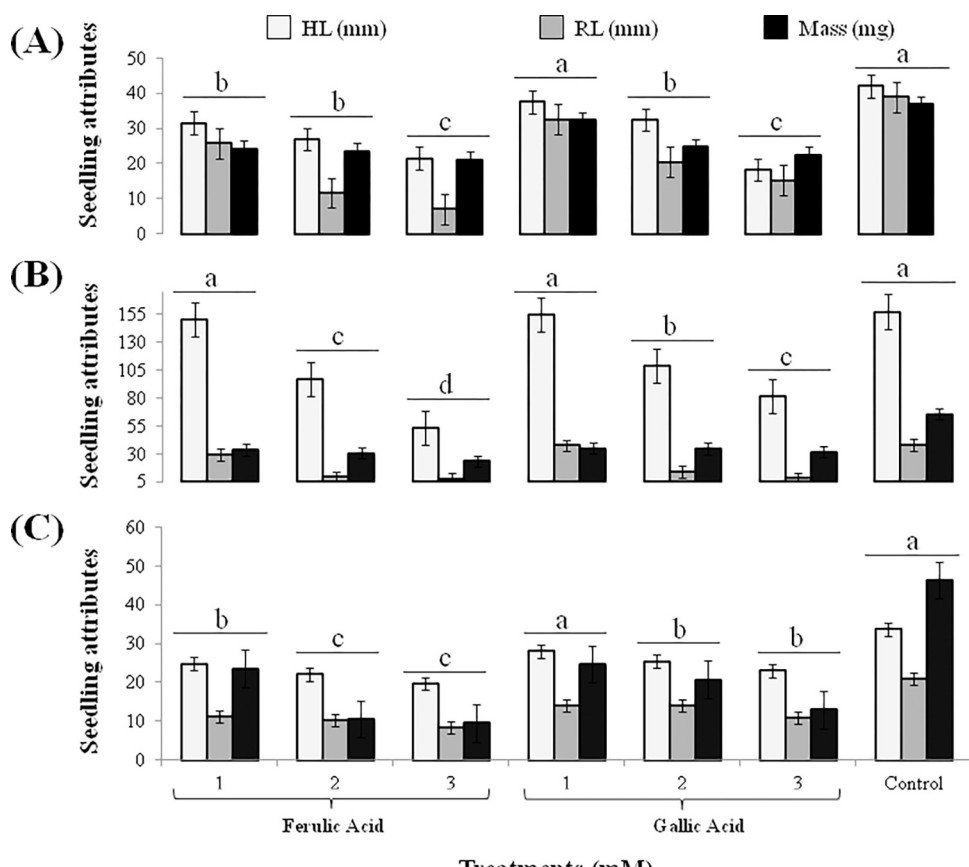

**Fig 2. Seedling attributes i.e., hypocotyl length (HL), radicle length (RL) and mass of weeds after treatment with 1, 2 and 3 mM ferulic acid (FA) and gallic acid (GA).** (A) effect on *Sinapis arvensis*; (B) effect on *Lolium multiflorum*; (C) effect on *Parthenium hysterophorus*. Values are presented as mean ± SE (n = 10). Bar values with different letters differ significantly from each other at $P < 0.05$.

to control. Similar results were obtained when protein contents of the weeds were estimated, whereby up to 40 to 60% protein contents were decreased with 2 and 3 mM FA and GA concentration, While, 1 mM concentration did not show any significant effect in comparison to the control (Fig 3).

## Oxidative damage induced in weeds by FA and GA

Level of oxidative stress induced by FA and GA, due to production of reactive oxygen species (ROS), in weeds was determined enzymatically and non-enzymatically. Catalase (CAT), superoxide dismutase (SOD) and peroxidase (POD) activities drastically increased up to 70 to 100% in comparison to control seedlings of tested weeds in response to 2 and 3 mM concentration of FA and GA. Level of lipid peroxidation, as determined by malondialdehyde (MDA), was also significantly elevated up to 60 to 80% in test weeds (Fig 4).

## Expression analysis of photosynthetic gene (psbA) of weeds in response to FA and GA

Treatment of weeds with FA and GA altered psbA gene expression levels in *S. arvensis*, *L. multiflorum* and *P. hysterophorus*. Differential psbA expression was identified in comparison to

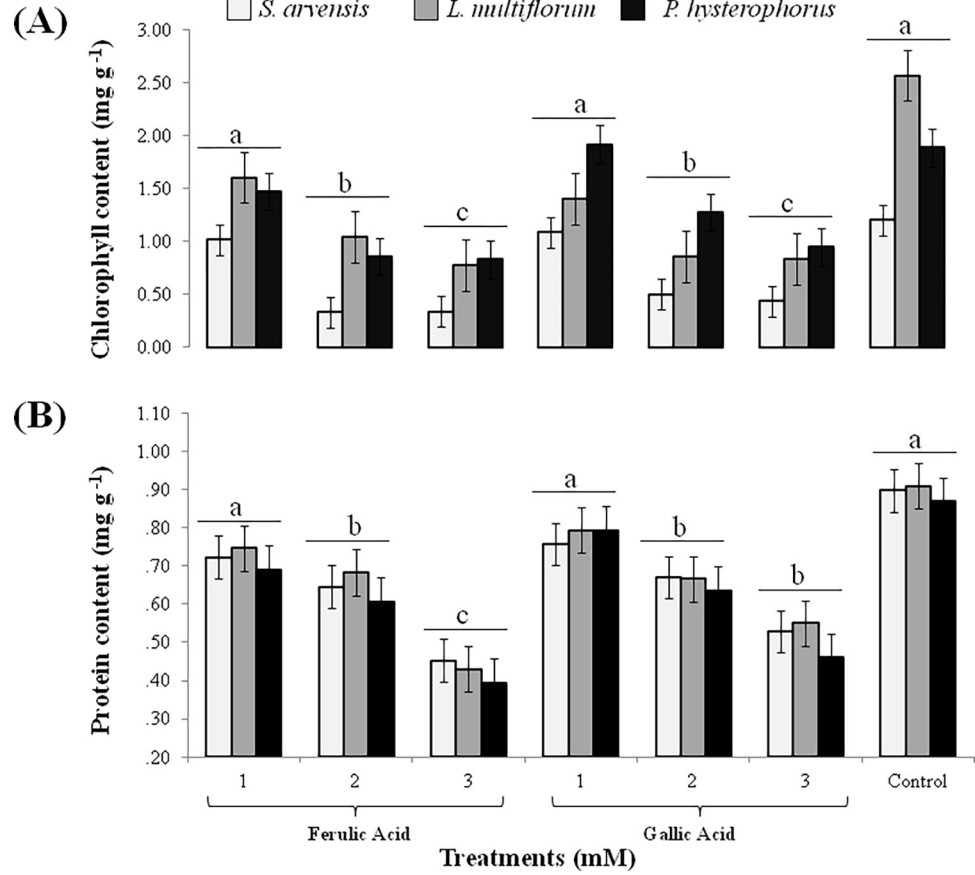

**Fig 3. Total chlorophyll and total protein contents of weeds in response to 1, 2 and 3 mM FA and GA.** (A) chlorophyll content; (B) protein content. Values are presented as mean ± SE (n = 10). Bar values with different letters differ significantly from each other at $P < 0.05$.

aqueous control treatment. psbA expression profile depicted its downregulation with increased concentrations of FA and GA (Fig 5A). All applied concentrations of FA i.e., 1, 2 and 3 mM had more negative effect in comparison to GA treatments. The levels of gene expression in weeds correlated with their chlorophyll contents. Highest inhibitory effect was demonstrated in *S. arvensis* (20 to 40 folds) followed by *L. multiflorum* (15 to 30 folds) and *P. hysterophorus* (2 to 10 folds). In control seedlings, psbA expression was relatively higher in all the treated weeds (Fig 5B).

## Molecular docking analysis of ferulic acid (FA) and gallic acid (GA) with D1 protein

To identify that whether phenolic acids bind to different receptors in D1 protein of weeds than those already known and targeted by PSII inhibitor herbicides and have developed resistance, *in silico* docking analysis of FA and GA with D1 protein was performed. Active sites of D1 protein were identified with SiteMap. Two binding sites in D1 were identified with FA with glide score = − 4.272. FA was predicted to establish one pi-pi stacking of its benzene ring with imadazole ring of histidine (his-198) and one hydrogen bond with threonine (thr-286) via hydroxyl group. GA showed two hydrogen bonds with thr-286 via two of its hydroxyl groups and glide score = − 5.055. FA, hence depicted more binding affinity with D1 protein then GA. Ligand-protein interactions are shown in Fig 6.

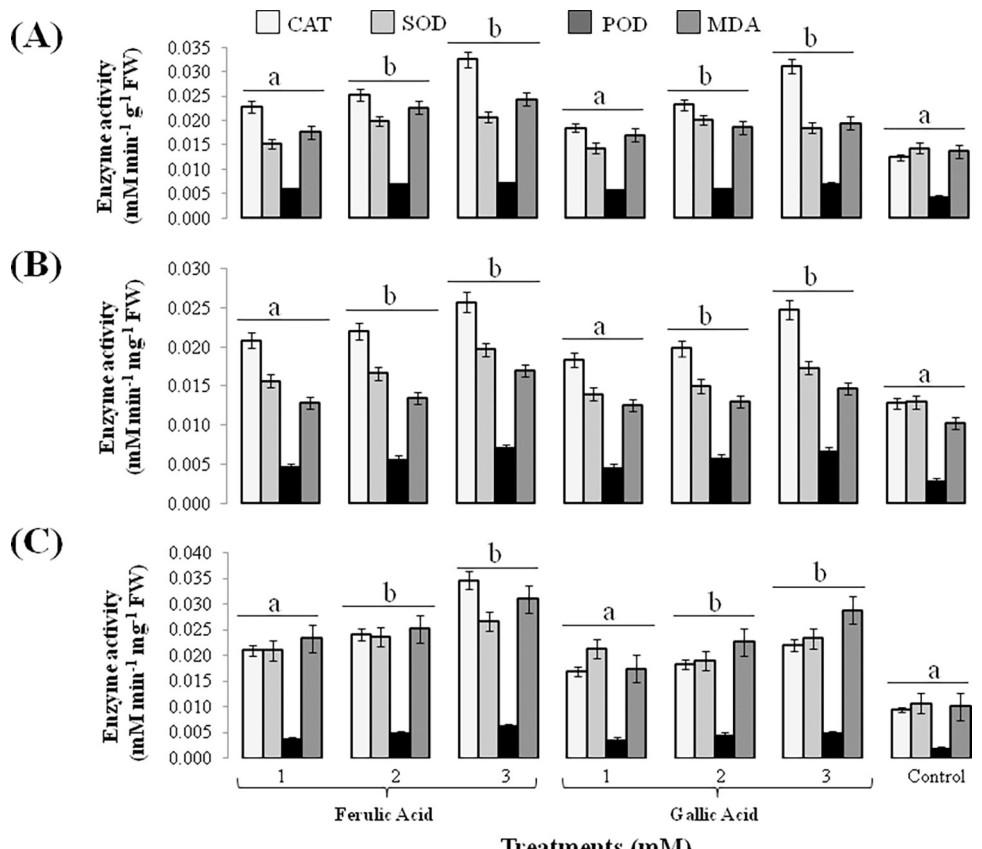

**Fig 4. Activity of antioxidant enzymes (CAT, SOD, POD) and lipid peroxidation (MDA) in response to 1, 2 and 3 mM FA and GA.** (A) effect on *Sinapis arvensis*; (B) effect on *Lolium multiflorum*; (C) effect on *Parthenium hysterophorus*. Values are presented as mean ± SE (n = 10). Bar values with different letters differ significantly from each other at $P < 0.05$.

## Comparative analysis of toxicity of ferulic acid and its analogs against germination and biochemical activity of weeds

With the advancement of bioinformatics and chemometric tools, new classes of herbicides can also be developed [52]. Based on structural chemistry of FA, twelve new compounds were designed in this study using OpenEye software. Molecular structures and formulae of analogs are given in Fig 7 and Table 1 respectively.

Analog 1, although did not inhibit germination percentage significantly and germination index of weeds, depicted its negative effect as germinated seeds either showed stunted growth or wilted within the time span of data collection. 0.8 mM and 1 mM FA also significantly reduced germination. Seeds grown with control treatments (0.25 mM DMSO and aqueous control) had no effect on germination (S3 and S4 Figs). Since, four (1, 6, 7 and 11) out of twelve analogs had inhibitory effect on both test species, the seedlings treated with these analogs were utilized for biochemical analysis. Levels of CAT, SOD and POD enzymes increased after 24 h of treatment (Fig 9). Significant increase in lipid peroxidation (MDA) was also observed with all tested analogs with highest effect induced by analog 11. For reference, 0.8 and 1 mM FA concentration was used which also altered enzymatic levels in tested weeds.

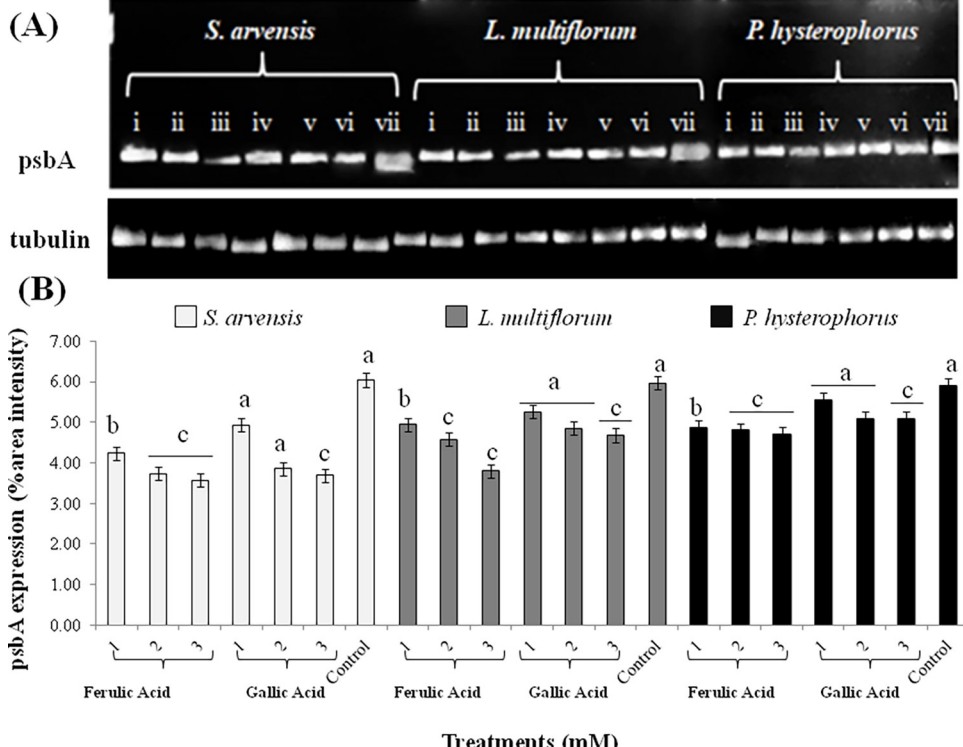

**Fig 5. Expression levels of psbA gene in weeds treated with different concentrations of ferulic acid (FA) and gallic acid (GA) as expressed with semi-quantitative PCR analysis.** (A) Downregulated transcription levels of psbA in *Sinapis arvensis*, *Lolium multiflorum* and *Parthenium hysterophorus* after 48 h of FA and GA treatment application in comparison to aqueous control. For each weed; (i) 1mM FA; (ii) 2mM FA; (iii) 3mM FA; (iv) 1mM GA; (v) 2mM GA; (vi) 3mM GA; (vii) aqueous control. (B) Relative expression of psbA gene in treated and control weed samples as depicted with relative percentage area intensity of bands. Values are presented as mean ± SE (n = 3). Bar values with different letters differ significantly from each other at $P < 0.05$.

## Discussion

Phenolic compounds, the secondary metabolites derived from plants, are recognized as antioxidants and bioactive agents for several therapeutic applications having antibacterial, anti-inflammatory, anti-cancerous and cardio-protective effects [53–55]. Phytochemical rich plant extracts can also be utilized as a source of eco-friendly herbicide development, as their allelochemicals including phenolic acids, flavonoids, alkaloids and terpenoids are known to suppress germination and growth of other plants [56, 57]. Mechanism of phytotoxicity of phenolics comprise of disruption of membrane integrity and repression of nutrients uptake, respiration, photosynthetic activity and protein synthesis of receptive plants [58]. Presumed significance of these compounds as growth suppressors might also be elucidated with their interference with active cell division in apical meristem of susceptible plant roots [17]. GA and FA, specifically, have been reported as potential phytotoxic compounds derived from more than 100 medicinal plants [59], against radicle and hypocotyl length of lettuce seedlings.

In this study, phytotoxic potential of FA and GA against prevalent herbicide-resistant weeds was evaluated. *In vitro* seed germination of *S. arvensis* reduced up to 40 to 60% and that of *L. multiflorum* diminished up to 60 to 80% in response to 0.8 and 1 mM FA respectively. Same concentrations of GA also inhibited up to 20 to 40% and 50 to 60% germination of *S. arvensis* and *L. multiflorum* respectively (Figs 1 and 2). Phenolic compounds have herbicidal properties since they are known to suppress germination and seedling growth of target plants

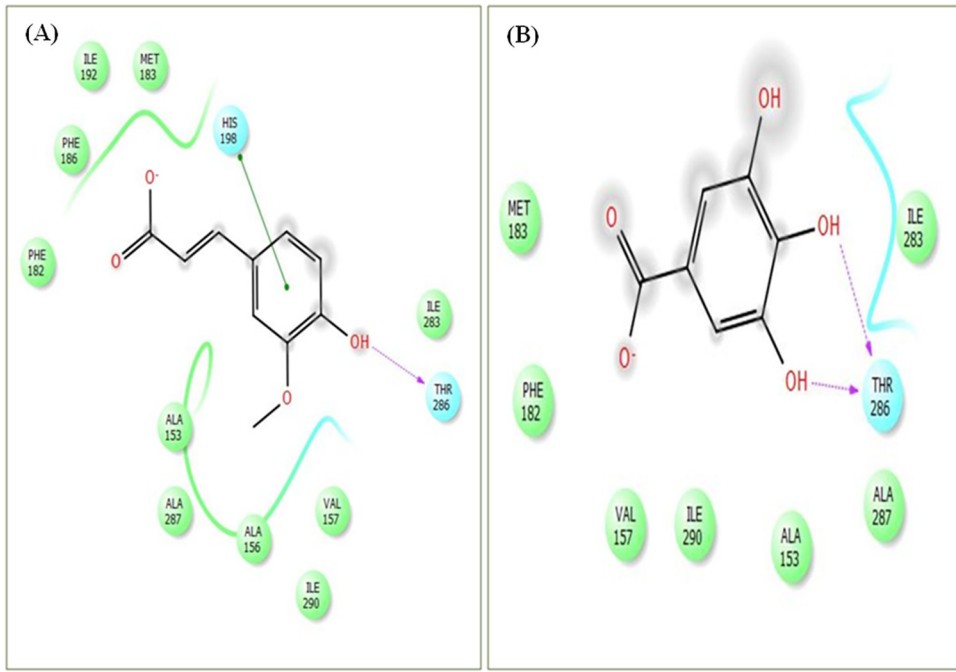

**Fig 6. Target binding sites in photosynthetic protein (D1) as identified by molecular docking analysis.** (A) docking with ferulic acid (FA), glide score = − 4.272; (B) docking with gallic acid (GA), glide score = − 5.055. Images were generated with PyMol. Higher binding affinity was shown by FA in comparison to GA.

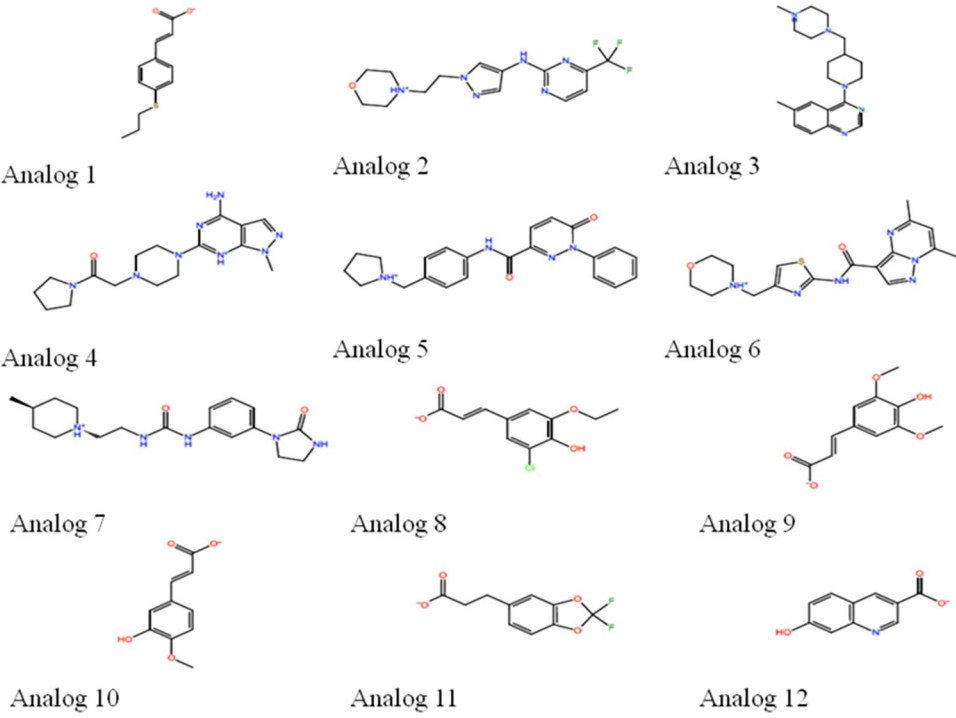

**Fig 7. Molecular structures of designed analogs.**

**Table 1. Molecular formulae of designed analogs with their labels.**

| Analog No. | Label | Molecular formula |
|---|---|---|
| Z235324677 | Analog 1 | 3-[4-(propylsulfanyl)phenyl]prop-2-enoic acid |
| Z1103016688 | Analog 2 | N-{1-[2-(morpholin-4-yl)ethyl]-1H-pyrazol-4-yl}-4-(trifluoromethyl)pyrimidin-2-amine |
| Z1576658419 | Analog 3 | 6-methyl-4-{4-[(4-methylpiperazin-1-yl)methyl]piperidin-1-yl}quinazoline, |
| Z1787515044 | Analog 4 | 2-(4-{4-amino-1-methyl-1H-pyrazolo[3,4-d]pyrimidin-6-yl}piperazin-1-yl)-1-(pyrrolidin-1-yl)ethan-1-one |
| Z339981218 | Analog 5 | 6-oxo-1-phenyl-N-{4-[(pyrrolidin-1-yl)methyl]phenyl}-1,6-dihydropyridazine-3-carboxamide |
| Z370422178 | Analog 6 | 5,7-dimethyl-N-{4-[(morpholin-4-yl)methyl]-1,3-thiazol-2-yl}pyrazolo[1,5-a]pyrimidine-3-carboxamide |
| Z433450950 | Analog 7 | 3-[2-(4-methylpiperidin-1-yl)ethyl]-1-[3-(2-oxoimidazolidin-1-yl)phenyl]urea |
| Z3219861126 | Analog 8 | (2E)-3-(3-hydroxy-4-methoxyphenyl)prop-2-enoic acid |
| Z3219955357 | Analog 9 | (2E)-3-(3-chloro-5-ethoxy-4-hydroxyphenyl)prop-2-enoic acid |
| Z3234818099 | Analog 10 | (2E)-3-(4-hydroxy-3,5-dimethoxyphenyl)prop-2-enoic acid |
| Z2469998588 | Analog 11 | 3-(2,2-difluoro-1,3-dioxaindan-5-yl) propanoic acid |
| Z2832012391 | Analog 12 | 7-hydroxyquinoline-3-carboxylic acid |

Analogs at lower concentration (0.1 mM) were not effective against seed germination of *S. arvensis* and *L. multiflorum* while at 0.5 mM, DMSO also caused germination inhibition, and therefore, 0.25 mM concentrations of analogs were used for germination assays. Relative to control treatments, i.e. DMSO and aqueous control, three analogs (6, 7 and 11) out of twelve, had significant effect on germination index (GI) and germination (G%) percentage of *S. arvensis* and *L. multiflorum* (Fig 8).

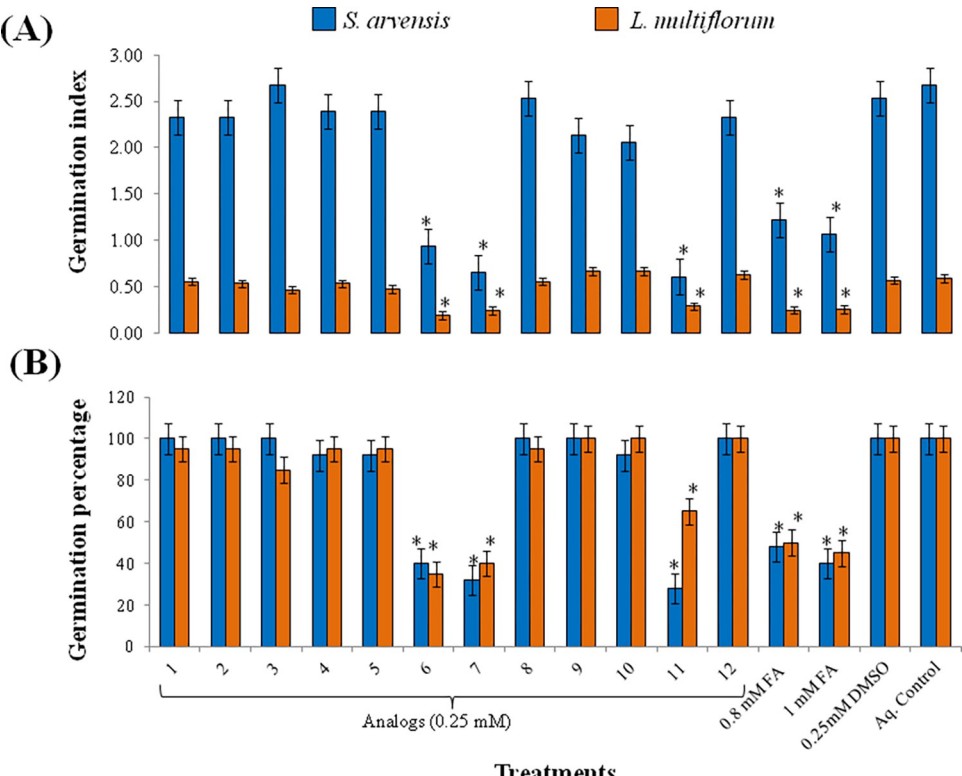

**Fig 8. Effect of analogs on *Sinapis arvensis* and *Lolium multiflorum* in response to 0.25 mM analogs, 0.8 and 1 mM ferulic acid (FA) and aqueous control.** (A) Germination index and (B) germination percentage. Analog 6, 7 and 11 had significant effect on GI and G% of both the weeds. Seed germination significantly reduced with FA at 0.8 and 1 mM concentration. Seeds grown with control treatments (0.25 mM DMSO and aqueous control) had no effect on germination. Data are presented as mean ± SE (n = 10). Bar values with asterisks differ significantly from solvent and aqueous control at $P < 0.05$.

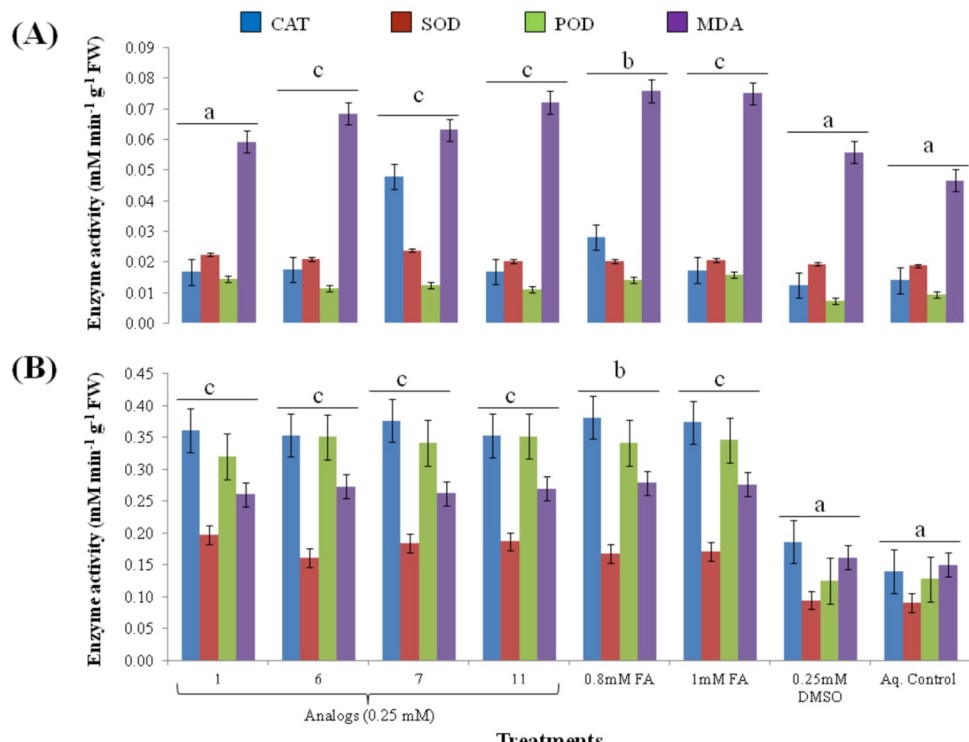

**Fig 9. Levels of antioxidant enzymes (CAT, SOD, POD) and lipid peroxidation (MDA) in response 0.25 mM analogs, 0.8 and 1 mM ferulic acid (FA), 0.25 mM DMSO and aqueous control.** (A) effect on *Sinapis arvensis*; (B) effect on *Lolium multiflorum*. Analogs induced significant elevation of enzymes and lipid peroxidation similar to 1 mM FA. Values are presented as mean ± SE (n = 3). Bar values with different letters differ significantly from each other at $P < 0.05$.

[42, 56, 57]. Phenolic acids, including syringic, 4-hydroxy-benzoic, vanillic cinnamic, 4-hydroxycinnamic and ferulic acids, have significant activity as weed inhibitors [60]. p-coumaric acid was reported to negatively affect the water uptake, maintenance of $Na^+/K^+$ channel and oxygen supply essential of germination [61]. This indicates that phenolic acids may also disrupt synthesis of important germination hormones. Furthermore, these allelochemicals have demonstrated to inhibit seed germination by restricting cellular respiration [62–64], thus changing cell membrane permeability [65, 66]. Studies also demonstrated reduced water uptake [67] accompanied with decreased enzymatic activity, caused by allelochemical stress induced in weed seeds during germination [68].

Allelochemicals can block the oxidation reaction in electron transport chain, thus inhibiting the photosynthetic activity of target plants [69]. Since total chlorophyll content (a and b) play considerable role in photosynthetic activity of plants [70], thus it is an important parameter to illustrate plant physiological condition after allelochemical application [7]. Present study indicated that the relative content of photosynthetic pigment (total chlorophyll content) of weeds significantly reduced in response to increased concentration of phenolic acids (Fig 3). Allelochemicals were also reported to efficiently restrain protein synthesis in cabbage seedlings [71]. The reduced chlorophyll content due to vanillic and p-hydroxy benzoic acids has been demonstrated [72]. Ferulic acid exhibited to have significant role in inhibition of chlorophyll and protein contents, accompanied with reduction of growth in tomato seedlings [73]. [56] suggested that decrease in photosynthetic pigment and activity is caused by reduction of transpiration

and stomatal conductivity in response to benzoic acid and cinnamic acid. FA has been proposed as an inhibitor of stomatal opening and photosynthetic rate in *Rhododendron delavayi* seedlings [57]. The FA and p-hydroxy benzoic acid acted as significant inhibitors of photosystem II in *Rumex acetosa* L. [74]. The oxygen evolving part of photosystem II is known to be the vulnerable part, whose mutilation significantly impacts the whole plant growth [40]. This elucidates the diminution of weed growth and biomass accompanied with decline of chlorophyll and protein contents.

Environmental stress produced by exogenously applied allelochemicals enhances reactive oxygen species (ROS) production in cells [75]. ROS activates free radicals that damage DNA, proteins and cell membranes, along with increasing levels of antioxidant enzymes and damage caused to cell by lipid peroxidation [76]. In present study, enzymatic and non-enzymatic activities predominantly increased (60 to 80%) as a result of FA and GA application in all the treated weeds (Fig 4). Generation of ROS in response to caffeic acid also resulted in altered activity of peroxidase (POD) while suppressing radicle and hypocotyl growth of mung bean [77]. Antioxidant enzymes (SOD, CAT, APX and GPX) elevated in fig leaf gourd roots as a result of cinnamic acid treatments that caused ROS accumulation [78]. Higher levels of MDA demonstrated in this study suggest increased lipid peroxidation in the weed cells, elucidating that initiation and accumulation of ROS is exceptionally damaging to cellular membranes and proteins [65].

Chloroplasts are most susceptible to oxidative stress and psbA (D1) protein, an essential part of PSII, is major damage target [66]. Disruption of this core protein may interrupt the oxygen production function of PSII [79]. Lipid peroxidation, resulted from ROS generation is one of the significant contributors that cause damage to PSII components including D1 protein [80]. High turnover rate of D1 is required for protection and integration of PSII [81]. Downregulation of psbA gene expression, in response to various molarities of FA and GA, specify their capability of disturbing photosynthetic activity of weeds owing to the induction of oxidative stress.

It is of substantial significance to comprehend the specific target sites of phenolic compounds to eventually elucidate and manipulate the physiological and biochemical effects depicted by them on weeds. *In silico* docking analysis of FA and GA indicated unexplored target sites, in D1 photosynthetic protein, as His-198 and Thr-286 to have strong binding affinity with them (Fig 6). Imidazole ring in histidine accounts for its aromatic attributes along with its major importance in several proteins [82], and therefore, its pi-pi stacking with FA could possibly inhibit function of D1 protein. Threonine, one of the hydroxyl containing amino acids, also plays significant role in phosphorylation of light harvesting complex of photosystem II, as threonine kinase, which in turn leads to balancing of light between the two photosystems [83–85]. Formation of one and two hydrogen bonds with FA and GA respectively, as depicted with docking analysis, predicts its plausible involvement in inhibition of photosynthetic activity and ultimately the weed growth. The two amino acids (his-198 and thr-286) depicting binding affinity with FA and GA, are different from the ones that are known targets of PSII inhibtor herbicides (ser-264 and his-215) and have reported to developed resistance in weeds [18, 36]. Prediction of unexplored sites of action with FA and GA thus provides opportunity to cope with continually increasing herbicide resistance in weeds by employing naturally derived compounds.

With the advancement of bioinformatics and chemometric tools, new classes of herbicides can also be developed [52]. Based on structural chemistry of ferulic acid, twelve new compounds were designed in this study using OpenEye software (Fig 7). Investigation of *in vitro* phytotoxic activity of these compounds against *S. arvensis* and *L. multiflorum* exhibited that 0.25 mM of four compounds was able to inhibit seed germination accompanied with oxidative

stress in them, in parallel to ferulic acid toxicity induce at 0.8 and 1mM (Fig 8). FA, having phenol group and propenoic acid side chain, easily forms phenoxy radical, thus accounting for its oxidative potential [86]. Among the analogs, side chain of FA was exchanged with esters and other active functional groups which would presumably enhance their toxicity against weeds. Esterification of ferulic acid (feruloyl ester) is known to enhance its antioxidant activity [87], while addition of amide group in it to form feruloyl amide enhances its activity even more [88]. Increased antioxidant activity of compounds is correlated to their phytotoxic potential [89]. Out of twelve designed analogs, analog six and seven (Fig 7, Table 1) showed significant effect on germination of *S. arvensis* and *L. multiflorum* and also elevated antioxidant enzyme levels and lipid peroxidation. Both the analogs had feruloyl amide side chains. Fluorinated analogs of FA are also employed during lignification process [90]. Analog 11 is also a fluorinated analog of ferulic acid (Fig 7, Table 1) and thus significantly diminished germination index while inducing oxidative damage to weeds. Further analyses via high throughput bio-informatic tools may help to identify specific receptors of designed analogs for advance studies.

## Conclusions

Present investigation implies that exogenously applied FA and GA are potentially phytotoxic (up to 3 mM concentration), against growth and development of *S. arvensis*, *L. multiflorum* and *P. hysterophorus* under soil condition. This study contributes to the understanding that investigating phytotoxic activity under rhizosphere environment is crucial to recognize interaction of phenolic compounds with prevalent herbicide-resistant weeds, for development of biologically derived herbicides, their possible mode of actions and primarily, their accurate doses as allelopathins. Identification of specific unexplored receptors of phenolic compounds may further provide foundation for targeted manipulation of such naturally derived compounds. Perception on inhibitory activity of analogs will present rationale for selecting organic molecules for developing feasible 'green herbicides' with novel target sites.

## Supporting information

**S1 Fig. Germination of *Sinapis arvensis* in response to ferulic acid (FA) and gallic acid (GA) treatments. (a)** 0.6 mM FA; **(b)** 0.8 mM FA; **(c)** 1 mM FA; **(d)** 0.6 mM GA; **(e)** 0.8 mM GA; **(f)** 1 mM GA and **(g)** Aqueous control. Seed germination was significantly inhibited with 0.8 and 1 mM FA and GA.
(TIF)

**S2 Fig. Germination of *Lolium multiflorum* in response to ferulic acid (FA) and gallic acid (GA) treatments. (a)** 0.6 mM FA; **(b)** 0.8 mM FA; **(c)** 1 mM FA; **(d)** 0.6 mM GA; **(e)** 0.8 mM GA; **(f)** 1 mM GA and **(g)** Aqueous control. Seed germination was significantly inhibited with 0.8 and 1 mM FA and GA.
(TIF)

**S3 Fig. Germination of *Sinapis arvensis* after treatments with analogs, ferulic acid (FA) and aqueous control. (a)** to **(l)** represent analog 1 to 12 (0.25 mM); **(m)** 0.25 mM DMSO; **(n)** aqueous control. Analog 1, 6, 7 and 11 reduced seed germination and wilted seedling, if grown.
(TIF)

**S4 Fig. Germination of *Lolium multiflorum* after treatments with analogs, ferulic acid (FA) and aqueous control. (a)** to **(l)** represent analog 1 to 12 (0.25 mM); **(m)** 0.25 mM DMSO; **(n)** aqueous control. Analog 1, 6, 7 and 11 reduced seed germination and wilted seedling, if

grown.
(TIF)

**S1 Raw image. Raw gel image for psbA and tubulin expression in *Sinapis arvensis*, *Lolium multiflorum* and *Parthenium hysterophorus* after 48 h in response to FA and GA treatments.** For each weed; (i) 1mM FA; (ii) 2mM FA; (iii) 3mM FA; (iv) 1mM GA; (v) 2mM GA; (vi) 3mM GA; (vii) aqueous control. (L = 1 kb DNA ladder; X = lane not shown in original manuscript). Gels were visualized and photographed for analysis in gel documentation system (Genosens 1560).
(TIF)

**S1 Table. Quantification data of reference gene (tubulin) determined through Image J software.**
(DOCX)

## Author Contributions

**Conceptualization:** Zahid Ali.

**Formal analysis:** Zahid Ali.

**Funding acquisition:** Zahid Ali.

**Investigation:** Sobia Anwar, Saadia Naseem.

**Supervision:** Saadia Naseem, Zahid Ali.

**Validation:** Sobia Anwar, Saadia Naseem, Zahid Ali.

**Visualization:** Saadia Naseem.

**Writing – original draft:** Sobia Anwar, Saadia Naseem.

**Writing – review & editing:** Zahid Ali.

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
