## [Decision Letter · Decision Letter 0]

10 Nov 2022

PONE-D-22-28844Biochemical analysis, photosynthetic gene (psbA) down–regulation, and in silico receptor prediction in weeds in response to exogenous application of phenolic acids and their analogsPLOS ONE

Dear Dr. Ali,

Thank you for submitting your manuscript to PLOS ONE. After careful consideration, we feel that it has merit but does not fully meet PLOS ONE’s publication criteria as it currently stands. Therefore, we invite you to submit a revised version of the manuscript that addresses the points raised during the review process.

We look forward to receiving your revised manuscript.

Kind regards,

Mayank Gururani

Academic Editor

PLOS ONE

“YES- Pakistan Science Foundation provided the resesrch funding through grant no. PSF-NSLP 663.”

Please respond by return e-mail so that we can amend your financial disclosure and competing interests on your behalf.

“No authors have competing interests.”

5. Please upload a copy of Figures S1 and S2, to which you refer in your text on page 13. If the figure is no longer to be included as part of the submission please remove all reference to it within the text.

Reviewers' comments:

Reviewer's Responses to Questions

**Comments to the Author**

1. Is the manuscript technically sound, and do the data support the conclusions?

Reviewer #1: Yes

Reviewer #2: Partly

2. Has the statistical analysis been performed appropriately and rigorously? 

Reviewer #1: I Don't Know

Reviewer #2: Yes

3. Have the authors made all data underlying the findings in their manuscript fully available?

Reviewer #1: No

Reviewer #2: Yes

4. Is the manuscript presented in an intelligible fashion and written in standard English?

Reviewer #1: Yes

Reviewer #2: Yes

5. Review Comments to the Author

Reviewer #1: The authors studied the efficacy of the ferulic acid and gallic acid to three prevalent herbicide-resistant weed species. In silico analysis found new receptors of ferulic acid and gallic acid in psbA protein. They also found four analogs for developing plant based formulation to suppress weed growth. However, there several items shoul be addressed.

1. I can not read the supporting information.

2. Figure 9 B, it is strange to me that the effect of analog 1 is a, it seems similar with analog 6,7 and 11.

3. The effect of ferulic acid, gallic acid and the analogs to the germination of P. hysterophorus is missing.

4. Figure 8, There is no effect of of analog 1 to the germiantion of weed species. So, not four but three analogs have been identified.

Reviewer #2: In general the manuscript is sound and the data are well presented! Based on data presented the conclusions are well sustained! Hence, I recommend publication of the manuscript after a minor revision!

The choice of a semiquantitative PCR technique for determination of psbA transcripts is not ideal! A Real-time RT-PCR approach would have been more appropriate and is recommended. However if this is not possible please include in manuscript or supporting materials the quantitation of the reference gene. It is really important that the expression of this gene should not be influenced by your treatments.

The manuscript should also be carefully read for minor language and syntax errors!

6. PLOS authors have the option to publish the peer review history of their article (what does this mean?). If published, this will include your full peer review and any attached files.

Reviewer #1: No

Reviewer #2: No

---

## [Author Response · Author response to Decision Letter 0]

2 Dec 2022

Response to Reviewers

On behalf of all authors, I am highly thankful to the reviewers for their valuable time they have given to improve this manuscript. We have thoroughly reviewed the comments and improved the manuscript accordingly. Below are the reviewer’s comments (black marked) and responses to the comments (red marked)

Response 1. Manuscript has been checked thoroughly and we followed the journal format requirements.

“YES- Pakistan Science Foundation provided the research funding through grant no. PSF-NSLP 663.”

Please respond by return e-mail so that we can amend your financial disclosure and competing interests on your behalf.

Response 2. The mentioned statement about financial disclosure is correct and is also added in the manuscript. Please see page 24, line 513 – 516.

“No authors have competing interests.”

Response 3. "The authors have declared that no competing interests exist." also added in the main manuscript. Please see page 24, line 511.

Response 4. Raw files of gel images and data are uploaded as supporting information. Captions for supporting information are added at the end of manuscript. Please see page 35, line 778 – 800. 

5. Please upload a copy of Figures S1 and S2, to which you refer in your text on page 13. If the figure is no longer to be included as part of the submission please remove all reference to it within the text.

Response 5. Fig S1 – S4 are uploaded individually as supporting information. Captions for all supplementary figures are added. Please see page 35, line 778 – 800. 

Response 6. Needful done. Reference list has been reviewed thoroughly and it is according to the Journal guidelines. There are no retracted articles cited in the manuscript. 

Reviewer #1: The authors studied the efficacy of the ferulic acid and gallic acid to three prevalent herbicide-resistant weed species. In silico analysis found new receptors of ferulic acid and gallic acid in psbA protein. They also found four analogs for developing plant based formulation to suppress weed growth. However, there several items should be addressed.

1. I cannot read the supporting information.

Response 1. Needful done. Supporting information including Fig S1 – S4 have been upload individually. Captions for all the supporting information are added at the end of the revised manuscript. Please see page 35, line 778 – 800. 

2. Figure 9 B, it is strange to me that the effect of analog 1 is a, it seems similar with analog 6,7 and 11.

Response 2. The valuable comment is highly acknowledged. The figure has been rechecked and corrected. Please see updated Fig 9. 

3. The effect of ferulic acid, gallic acid and the analogs to the germination of P. hysterophorus is missing.

Response 3. In vitro phytotoxicity of compounds was investigated against S. arvensis and L. multiflorum as representatives of dicot and monocot species. These two species grow easily and rapidly under similar in vitro conditions. [Please see: 

• Javaid, M. M., Mahmood, A., Alshaya, D. S., AlKahtani, M. D., Waheed, H., Wasaya, A., ... & Fiaz, S. (2022). Influence of environmental factors on seed germination and seedling characteristics of perennial ryegrass (Lolium perenne L.). Scientific Reports, 12(1), 1-11. 

• Singh, A., Mahajan, G., & Chauhan, B. S. (2022). Germination ecology of wild mustard (Sinapis arvensis) and its implications for weed management. Weed Science, 70(1), 103-111.] 

In our earlier research it was observed that germination of S. arvensis and L. multiflorum took up to 3 and 5 days respectively while P. hysterophorus germinated between 10 to 15 days. 

[Please see: Anwar, S., Naseem, S., Karimi, S., Asi, M. R., Akrem, A., & Ali, Z. (2021). Bioherbicidal Activity and Metabolic Profiling of Potent Allelopathic Plant Fractions Against Major Weeds of Wheat—Way Forward to Lower the Risk of Synthetic Herbicides. Frontiers in plant science, 333.] 

Thus preliminary phytotoxicity and dose optimization of phenolic acids and analogs was performed against one dicot and monocot species each to determine their earlier responses for the purpose of standardization of experiments. 

4. Figure 8, There is no effect of analog 1 to the germination of weed species. So, not four but three analogs have been identified.

Response 4. In principle we agree with this statement. Although analog 1 did not show significant inhibitory effect on the germination percentage but it was observed to have negative impact on seedling growth, since the seeds germinated, but could not grow well and wilted within the time of data collection. Statements have been amended as per suggestion by the worthy reviewer in the manuscript. Please see page 18, 19; line 378, 391 – 393. Further, biochemical analysis of investigated analogs against weeds also depicted negative results, including analog 1. Please see line 395 – 398.

Reviewer #2: In general, the manuscript is sound and the data are well presented! Based on data presented the conclusions are well sustained! Hence, I recommend publication of the manuscript after a minor revision!

The choice of a semiquantitative PCR technique for determination of psbA transcripts is not ideal! A Real-time RT-PCR approach would have been more appropriate and is recommended. However if this is not possible please include in manuscript or supporting materials the quantitation of the reference gene. It is really important that the expression of this gene should not be influenced by your treatments.

The manuscript should also be carefully read for minor language and syntax errors!

We appreciate the comment. Unfortunately, real-time PCR could not be performed, but the semi-quantitative PCR was performed with ultimate care and replicated to obtain valid results. Quantification of reference gene (tubulin) was done. Its expression was found to be uninfluenced by application of treatments, which can be been seen by the banding patterns and their quantification. The table for tubulin quantification along with the analysis of psbA gene is added as supporting information. Please see S1 Table. The manuscript has been carefully proofread for minor errors.

---

## [Decision Letter · Decision Letter 1]

4 Jan 2023

Biochemical analysis, photosynthetic gene (psbA) down–regulation, and in silico receptor prediction in weeds in response to exogenous application of phenolic acids and their analogs

PONE-D-22-28844R1

Dear Dr. Ali,

We’re pleased to inform you that your manuscript has been judged scientifically suitable for publication and will be formally accepted for publication once it meets all outstanding technical requirements.

Kind regards,

Mayank Gururani

Academic Editor

PLOS ONE

Additional Editor Comments (optional):

Reviewers' comments:

Reviewer's Responses to Questions

**Comments to the Author**

1. If the authors have adequately addressed your comments raised in a previous round of review and you feel that this manuscript is now acceptable for publication, you may indicate that here to bypass the “Comments to the Author” section, enter your conflict of interest statement in the “Confidential to Editor” section, and submit your "Accept" recommendation.

Reviewer #2: All comments have been addressed

2. Is the manuscript technically sound, and do the data support the conclusions?

Reviewer #2: Yes

3. Has the statistical analysis been performed appropriately and rigorously? 

Reviewer #2: Yes

4. Have the authors made all data underlying the findings in their manuscript fully available?

Reviewer #2: Yes

5. Is the manuscript presented in an intelligible fashion and written in standard English?

Reviewer #2: Yes

6. Review Comments to the Author

Reviewer #2: Thank you for considering my comments!

Given the authors response I consider the manuscript acceptable!

7. PLOS authors have the option to publish the peer review history of their article (what does this mean?). If published, this will include your full peer review and any attached files.

Reviewer #2: No

---

## [Editor Report · Acceptance letter]

8 Jan 2023

PONE-D-22-28844R1 

Biochemical analysis, photosynthetic gene (psbA) down–regulation, and *in silico* receptor prediction in weeds in response to exogenous application of phenolic acids and their analogs 

Dear Dr. Ali:

I'm pleased to inform you that your manuscript has been deemed suitable for publication in PLOS ONE. Congratulations! Your manuscript is now with our production department. 

Kind regards, 

on behalf of

Dr. Mayank Gururani 

Academic Editor

PLOS ONE